# Evaluation of β-Lactamase Enzyme Activity in Outer Membrane Vesicles (OMVs) Isolated from Extended Spectrum β-Lactamase (ESBL) *Salmonella* Infantis Strains

**DOI:** 10.3390/antibiotics12040744

**Published:** 2023-04-13

**Authors:** Valeria Toppi, Gabriele Scattini, Laura Musa, Valentina Stefanetti, Luisa Pascucci, Elisabetta Chiaradia, Alessia Tognoloni, Stefano Giovagnoli, Maria Pia Franciosini, Raffaella Branciari, Patrizia Casagrande Proietti

**Affiliations:** 1Department of Veterinary Medicine, University of Perugia, Via S. Costanzo 4, 06126 Perugia, Italy; 2Department of Pharmaceutical Sciences, University of Perugia, Via del Liceo 1, 06123 Perugia, Italy

**Keywords:** outer membrane vesicles, β-lactamase, *Salmonella* Infantis, nitrocefin assay, antimicrobial-resistance

## Abstract

Outer membrane vesicles (OMVs) are nanoparticles released by Gram-negative bacteria, which contain different cargo molecules and mediate several biological processes. Recent studies have shown that OMVs are involved in antibiotic-resistance (AR) mechanisms by including β-lactamase enzymes in their lumen. Since no studies have as yet been conducted on *Salmonella enterica* subs. *enterica* serovar Infantis’ OMVs, the aim of the work was to collect OMVs from five *S*. Infantis β-lactam resistant strains isolated from a broiler meat production chain and to investigate whether β-lactamase enzymes are included in OMVs during their biogenesis. OMVs were isolated by means of ultrafiltration and a Nitrocefin assay quantified the presence of β-lactamase enzymes in the OMVs. Transmission electron microscopy (TEM) and dynamic light scattering (DLS) were used to identify the OMVs. The results showed that all strains release spherical OMVs, ranging from 60 to 230 nm. The Nitrocefin assay highlighted the presence of β-lactamase enzymes within the OMVs. This suggests that β-lactamase enzymes also get packaged into OMVs from bacterial periplasm during OMV biogenesis. An investigation into the possible role played by OMVs in AR mechanisms would open the door for an opportunity to develop new, therapeutic strategies.

## 1. Introduction

Outer membrane vesicles (OMVs) are spherically bilayered nanoparticles, ranging in size from 10 to 300 nm, released into the extracellular milieu from the Outer Membrane (OM) of Gram-negative bacteria [1]. OMVs biogenesis is the result of a budding process and therefore three different main mechanisms have been proposed based on several biochemical and genetic studies. The first process involves the dissociation of the outer membrane in specific zones lacking proper attachments to underlying structures (e.g., peptidoglycan); the second is due to the presence of misfolded proteins, which accumulate in nanoterritories where crosslinks between peptidoglycan and other components of the bacterial envelope are either locally depleted or displaced; and the third, in which some changes in LPS composition also modulate OMV biogenesis, presumably by generating a differential curvature, fluidity, and/or charge in the outer membrane [2]. Several bacterial species have been shown to produce OMVs, e.g., *Escherichia coli* [1,3], *Pseudomonas aeruginosa* [4], *Campylobacter jejuni* [5,6], *Borrelia burgdorferi* [7,8], *Shigella* spp. [9], *Helicobacter pylori* [10], *Vibrio* spp. [11] *Neisseria* spp. [12], and *Salmonella* spp. [2].

Since OMVs arise from the OM of Gram-negative bacteria, they contain several outer membrane components as well as a soluble content inside and attached to their Outer Vesicle surface. The composition of OMVs cargo is modulated by growth conditions and is also highly influenced by the interactions between host cells and bacteria [13]. For instance, OMVs enclose in their lumen different types of molecules, such as proteins, LPS, lipids, metabolites, virulence factors, and also nucleic acids, derived from the bacterial periplasm that is located between the inner membrane and outer membrane [1]. Based on the nature of the cargo, OMVs are involved in multiple processes, including intracellular and extracellular communication, quorum sensing, horizontal gene transfer, interbacterial killing, and inter- and intra-species delivery of molecules, e.g., toxins and virulence factors. Moreover, OMVs are also involved in stress responses, such as biofilm formation, and participate in modulating the host immune response [1,2,11,12,13]. In addition to their role in bacterial communication, the transfer of virulence factors as cargoes to OMVs enhance bacterial survival in the host. In fact, OMVs are secreted as distinct entities which mediate the transfer of virulence factors, adhesion molecules, toxins, and immunomodulatory compounds. In particular, many virulence factors are secreted into OMVs, such as the vacuolating toxin VacA of *Helicobacter pylori* and the HlyE hemolysin in *Salmonella enterica* serovar Typhi [2]. In this way, they constitute a separate secretory system, which operates in Gram-negative bacteria to gain access to host tissues and the bloodstream. OMV production under in vitro conditions has been observed during bacterial growth on solid and liquid media, in biofilms, and also during intracellular infections [13].

Recent evidence indicates that OMVs, in addition to their multiple roles, are able to protect bacteria not only from phages and environmental stress factors but also from antibiotics [14].

It has been demonstrated that OMVs play a key role in antibiotic-resistance mechanisms by disseminating genetic determinants of antibiotic-resistance [14].

Recent studies have pointed out that OMVs can provide immediate protection for bacteria, acting as decoys that bind or absorb antibiotics and toxins, before the molecules reach their targets on bacteria [2]. Moreover, knowing that bacteria in biofilms are more resistant to the actions of antibiotics than their planktonic counterparts, a recent proteomic analysis obtained from planktonic growth and biofilm in *P. aeruginosa* revealed that the drug-binding proteins were more concentrated in biofilm OMVs [15].

OMVs can also carry enzymes that mediate antibiotic protection, such as β-lactamase enzymes. These enzymes are located in the bacterial periplasm, where they inactivate the β-lactam molecules before they reach their targets situated on the cytoplasmic membrane, the penicillin-binding proteins. The origin of extracellular β-lactamase activity was initially thought to be exclusively from lysed and broken cells. As observed in *P. aeruginosa*, it seems that the enzyme is packaged into OM vesicles. In fact, Ciofu et al. [15], using polyclonal antibodies against chromosomal β-lactamase of *Pseudomonas aeruginosa*, demonstrated the presence of β-lactamase in OMVs isolated from three β-lactam-resistant clinical isolates and one β-lactam-sensitive strain of the bacterium [15]. There is evidence to indicate that vesicles carrying β-lactamases protect not only the producer cell but also some other bacteria co-present in the micro-environment [16,17]. Indeed, it has been demonstrated that the β-lactamase associated with OMVs produced by some members of *Bacteroides* spp. can protect some commensal and enteric pathogens, such as *Salmonella* Typhimurium from the cefotaxime, a third-generation cephalosporin. In addition, *bla*Z, a lactamase protein, was found to be associated with the vesicles produced by *Staphylococcus aureus*, a Gram-positive bacterium [18].

*Salmonella* Infantis is a poultry-adapted *Salmonella enterica* serovar that is increasingly reported in broilers and is also regularly identified among human salmonellosis cases.

Over recent years, the increasing incidence of *S.* Infantis infection in humans and animals has also been complicated by the spread of multidrug-resistant (MDR) and extended spectrum β-lactamase (ESBL) clones in several European countries, including Switzerland [19], Slovenia [20], Hungary, Austria, Poland [21], Israel [22], Germany [23], and Italy [23,24]. *S.* Infantis ESBL strains mediate resistance to third-generation cephalosporins [17]. Recently, resistance to extended-spectrum cephalosporins such as cefotaxime, ceftriaxone, ceftazidime, and ceftizoxime, which are mainly produced by β-lactamase, is increasing. This poses serious public health implications, especially because these two classes are critically important human antimicrobials being the drug of choice for human salmonellosis.

Since there are no scientific studies to date on *S.* Infantis’ Outer Membrane Vesicles the aim of the research was to isolate and identify OMVs from MDR-ESBL *S.* Infantis strains. Furthermore, in order to study their involvement in antibiotic-resistance mechanisms, the possible inclusion and quantification of β-lactamase enzymes in OMVs were also evaluated.

## 2. Results

### 2.1. Transmission Electron Microscopy (TEM)

OMVs isolated from the MDR and ESBL *S.* Infantis cultures were spherical in shape and ranged from 60 to 230 nm in size. OMVs appeared as single vesicles or they were organized in clusters and exhibited an electrodense appearance (Figure 1).

### 2.2. Dynamic Light Scattering (DLS) Analysis

Photocorrelation analysis highlighted the presence of two populations at approximately 50 and 150 nm in both OMV samples. No major differences were observed, which suggested consistent features among the different samples. The analysis corresponded with the TEM observations, where outer membrane vesicles between 50 and 200 nm were observed (Figure 2).

### 2.3. Evaluation of β-Lactamase Enzymatic Activity

The β-lactamase activity values measured in the OMV concentrates, the eluates, and the filtrates normalized to the protein content are reported in Appendix A.

The β-lactamase enzymatic activity in OMV concentrates showed significantly higher values when compared with 0.45 filtrate (*p* < 0.01) and eluted samples (*p* < 0.05), respectively. No statistically significant differences were observed between the 0.45 filtrate and eluted samples (Figure 3).

In three strains (1,2,5), the value of β-lactamase enzymes was higher in the OMV concentrate (515.13, 554.14, and 1905.47 mU/mg, respectively) than in both the eluted (157.13, 289.34 and 118.70) and the 0.45 μm filtrate samples (192.51; 175.43 and 206.74 mU/mg). In two strains (3,4), the β-lactamase activity value was lower in the OMV concentrate (524.60 and 255.38) than in both the eluted (634.22 and 400.57 mU/mg) and the 0.45 μm filtrate samples (471.94 and 211.54 mU/mg).

## 3. Discussion

Bacterial OMVs are an intelligent strategy of mediation and exchange between intra- and interspecific micro-organisms in the micro-environment. They are involved in several processes, including protection from antibiotics and horizontal gene transfer. More specifically, OMVs mediate AR in various ways, as reported in the literature [25].

In fact, several studies have demonstrated that OMVs enable antibiotic resistance, including antibiotic-inactivating enzymes such as β-lactamase enzymes in their lumen [26,27].

Furthermore, it has been reported that OMVs from *Porphyromonas gingivalis* are able to confer resistance to antibiotics without the presence of resistance enzymes, by absorbing Chlorhexidine themselves [13].

Chattopadhyay et al. [14] observed the protecting action against antibiotics of OMVs isolated from the Antarctic bacterium *Pseudomonas syringae*. In fact, the authors demonstrated that the growth inhibitory effect of colistin and melittin, two membrane-active antibiotics, was reversed by the addition of outer membrane vesicles from the same organism to the culture medium. In this way, it was clear that the OMVs protect the bacteria by binding and removing the antimicrobial peptides from the extracellular milieu [14].

Moreover, it has been demonstrated that OMVs actively participate in disseminating the genes responsible for antibiotic resistance. Indeed, OMVs use horizontal gene transfer to mediate the acquisition of antibiotic-resistance gene determinants by those susceptible bacteria that are not only co-specific but also share the same habitat [14,26].

In our study, we considered five *S.* Infantis strains, previously identified as multi-drug resistant (MDR) and ESBL producers. *S.* Infantis strains were resistant to tetracycline, sulfamethoxazole/trimethoprim, and nalidixic acid, thus confirming the typical pattern of multi-resistance of the European *S.* Infantis clone [22]. To date, no studies have reported on the isolation and identification of OMVs from *S*. Infantis. We isolated OMVs from all strains of *S.* Infantis examined (*n* = 5). TEM analysis revealed that *S*. Infantis’ vesicles are spherical with a diameter ranging from 60 to 230 nm, in accordance with the previously published data on OMVs produced by *E. coli* [28], *Klebsiella pneumoniae* [29] and *Pseudomonas aeruginosa* [30].

The morphology was also similar to OMVs isolated by other *S*. *enterica* serovars, such as *S*. Choleraesuis and *S.* Typhimurium [31].

These observations were confirmed by DLS analysis which revealed two main populations at 50 and 150 nm in all OMV samples tested.

In this work, we measured the β-lactamase enzyme activity to understand whether β-lactamase enzymes are included in vesicles during their biogenesis.

Our results demonstrated that there was a statistically significant difference between the value measured within the concentrate with OMVs and that detected in the filtrate and eluate. Instead, no statistically significant difference was observed between filtrated and eluted samples. The activity value detected within the filtrate samples may be due to either the presence of free β-lactamases or the presence of OMVs.

Although Kim et al. [27] showed that Nitrocefin can enter OMVs through porins, sonication enables OMVs to be lysed and thus release all the β-lactamase present in their lumen. In our case, the β-lactamase activity values related to the 0.45 μm filtrate samples may have been underestimated, since filtrate samples were not subjected to sonication, contrary to the OMV concentrate samples.

In the concentrates, OMVs are retained by the Amicon filter with the 100-kDa cut-off, and for this reason, the β-lactamase activity value detected in these samples can only be attributed to the presence of β-lactamases in the vesicle lumen. On the contrary, the β-lactamase activity value detected in the eluate is due only to free β-lactamase which is not retained by the 100-kDa cut-off filter, as it weighs between 30 and 40 kDa. This is the first study in which the detection of β-lactamase with the Nitrocefin assay is used on 0.45 μm filtrate, eluate, and OMV concentrate samples.

With regards to the quantification of β-lactamase enzyme activity, it should be stressed that the value of β-lactamase enzymes in three strains (1,2,5) was higher in the OMV concentrate than in both the eluted and 0.45 μm filtrate samples. Instead, the value of the β-lactamase activity in two strains (3,4) was lower in the OMV concentrate than in both the eluted and 0.45 μm filtrate samples. This difference may be due to the diversity between the bacterial strains. A larger sample size would be useful to better investigate this phenomenon.

The values for β-lactamase enzyme activity obtained in our work are not currently comparable with other data due to the lack of similar studies in the literature. *S.* Infantis has emerged as the fourth most frequent serovar to cause human salmonellosis in Europe and is becoming a serious public health concern. *S.* Infantis is the serovar most frequently isolated in broiler flocks and the fourth most common in breeding flocks and laying hens in the European Union [28]. Casagrande Proietti et al. [27] have shown the spread and persistence of the ESBL strains of *S.* Infantis that are also resistant to cephalosporins, included in the list of critically important antimicrobials for human medicine by the WHO [31]. *S.* Infantis OMVs carrying β-lactamase enzymes could represent a strategy to protect resistant or susceptible bacteria co-present in the micro-environment by degrading β-lactam antibiotics before they can affect the bacteria, as observed in *E. coli* [26] and *Moraxella catarrhalis* [15].

Moreover, β-lactamases packed into the OMV lumen are relatively safe from dilution and degradation as a result as reported in the literature [31]. In fact, it has been reported that OMVs isolated from *Moraxella catarrhalis* were found to be able to protect the β-lactamase enzyme, produced by this bacterium, from the inactivation by anti-β-lactamase antibody, packaging the enzyme in their lumen [15].

In this scenario, *S*. Infantis’ OMVs containing β-lactamase enzymes could play a role in AR mechanisms.

## 4. Materials and Methods

### 4.1. Experimental Design

Five MDR and ESBL producers *S.* Infantis strains from a broiler meat production chain investigated in two previous studies [22,27] were selected in this work. Bacterial strains were stored at −80 °C until use. To evaluate the inclusion of β-lactamase enzymes in OMVs during their biogenesis, the β-lactamase activity value was measured in three different samples from three subsequent steps of the OMV isolation procedure. OMVs were isolated by means of 0.45 μm filtration and ultrafiltration. First, the β-lactamase enzyme activity in the 0.45 μm filtrate was tested, in which both vesicles and potential free β-lactamases were present. Then, β-lactamase enzyme activity was tested both in eluate and concentrate samples resulting from the ultrafiltration step.

### 4.2. Isolation of S. Infantis OMVs

In order to isolate the OMVs, all MDR and ESBL *S.* Infantis strains were cultured on Luria-Bertani (LB) agar plates (Thermo Fisher Scientific, city name, Milan MI, Italy) at 37 °C for 12/24 h and then on LB medium (Thermo Fisher Scientific, Milan MI, Italy) at 37 °C for 6–8 h with orbital shaking (220 rpm) under aerobic conditions.

The OMVs were prepared from liquid cultures of *S.* Infantis according to the method described in the literature with modifications [3,29]. OMVs were isolated from the late exponential phase of five *S.* Infantis strains, based on the previously published procedure [24,25,26]. Two separate tests were set up for each strain. A volume of 3 mL of overnight (O/N) culture was inoculated in 300 mL of LB medium (Thermo Fisher Scientific, Milan MI, Italy). The bacterial inoculum was cultured at 37 °C under orbital shaking (180 rpm) with aeration until OD_600_ reached 1.8 (up to 6–8 h of incubation). The cultures were centrifuged at 4500× *g* for 20 min at 4 °C in order to remove bacterial cells. A vacuum pump (Euroclone, Milano, Italy) filtered supernatants through a 0.45 μm filter membrane to remove the remaining bacteria and cell debris. The filtrate was checked for sterility by plating 100 μL on LB agar plates. The bacteria-free filtrates were concentrated with a 100 kDa membrane using the Amicon Ultrafiltration system (Merck Millipore, Billerica, MA, USA) to a volume of approximately 400 μL. The concentrated filtrates with OMVs were washed twice in a phosphate buffer (PBS pH 7.2). The OMV preparations were checked for the presence of remaining bacterial cells by plating the vesicle suspension on LB agar plates incubated at 37 °C O/N. OMVs were stored at −20 °C until used for experiments.

### 4.3. Transmission Electron Microscopy (TEM)

Transmission electron microscopy (TEM) was used to investigate OMV morphology. Ten microliters from each *S.* Infantis sample were adsorbed for 15 min on Formvar-coated copper grids (Electron Microscopy Sciences, Hatfield, PA, USA) in a humified chamber with the coated side towards the suspension. After the incubation, the grids were washed twice in 0.22 μm filtered PBS (0.1 M pH 7.3), followed by a final wash in distilled water. The excess liquid was removed with filter paper. The samples were contrasted with 2% uranyl-acetate dissolved in distilled water for 5 min. The grids were then washed twice in distilled water, air dried, and observed under a Philips EM208 transmission electron microscope equipped with a digital camera (CUMEF—University Centre of Electron and Fluorescence Microscopy, Perugia, Italy).

### 4.4. Dynamic Light Scattering (DLS)

The hydrodynamic size distribution of NVs was measured by dynamic light scattering (DLS) analysis, using a NICOMP 380 ZLS equipped with a 35 mW He-Ne Coherent Innova 70-3 laser source at 654.0 nm and APD detector (Particle Sizing System, Inc., Santa Barbara, CA, USA). Samples were analyzed after purification in 0.22 µm filtered PBS medium at 20 °C, as reported above.

### 4.5. Evaluation of β-Lactamase Activity

To investigate the presence of the β-lactamase enzymes in OMVs, β-lactamase activity was quantified by a colorimetric β-lactamase activity assay kit (β-Lactamase Activity Assay Kit, Sigma-Aldrich, Saint Louis, MO, USA) according to the manufacturer’s instructions. The β-lactamase activity was measured in three different samples: the filtered 0.45 μm supernatants, the eluted samples from ultrafiltration, and OMV concentrates. Two different tests were set up for each strain. The colorimetric assay is based on the hydrolysis of Nitrocefin, a chromogenic cephalosporin that produces a coloured, spectrophotometrically measurable product (OD_490_) proportional to the enzymatic activity present. A lyophilized, positive control included in the kit was used. The quantity of enzyme capable of hydrolysing 1.0 µM of Nitrocefin/min at 25 °C (pH 7.0) corresponds to 1 U of β-lactamase. To liberate β-lactamase from the OMV lumen, each OMV sample (30 μL) was sonicated for 5 min and cooled on ice for 5 min. Equal aliquots of OMVs, the eluted and the 0.45 μm filtrate (5 μL, 20 μL, and 20 μL, respectively) were dispensed into the wells of a clear, flat-bottomed 96-well plate (Thermo Fisher Scientific-Nunclon 96 flat bottom transparent polystyrene), and Nitrocefin and β-lactamase assay buffer (provided by the kit) were added to reach a final volume of 100 μL. The absorbance at OD_490_ was immediately measured in kinetic mode for 60 min at room temperature. Each sample was analysed in triplicate. A standard curve was generated using 0, 4, 8, 12, 16, and 20 nmol of Nitrocefin.

The protein content of the specific β-lactamase activity of each sample was normalized and expressed in milli-units/milligram of protein.

### 4.6. Protein Content Quantification

To determine the protein concentration, the 0.45 filtrate and eluted samples were previously centrifuged at 10,000× *g* for 10 min at 4 °C. The supernatant was discarded, and subsequently, the weight of the pellet was determined and it was resuspended in PBS (1 μL/mg of sample). The supernatants obtained after sonication and centrifugation at 4 °C for 13,500× *g* for 20 min were stored at −20 °C until use. OMV proteins were quantified using the Bradford assay, with bovine serum albumin (BSA) as the standard.

### 4.7. Statistical Analysis

Enzymatic activity data were presented as the mean ± standard deviation (SD). Significant differences in the enzymatic activity were determined by the Kruskal–Wallis test followed by Dunn’s post-hoc test (JASP Version 16.2). A *p*-value < 0.05 was considered significant for the analysis. All data were analyzed using GraphPad Prism (GraphPad, version 7, San Diego, CA, USA).

## 5. Conclusions

In this work, OMVs were isolated from five MDR and ESBL *S.* Infantis strains and the presence of β-lactamase enzyme activity was evaluated. It highlighted the presence of β-lactamase enzyme activity within the outer membrane vesicles, confirming the hypothesis that β-lactamase enzymes are packaged in OMVs’ lumen during their biogenesis.

However, further in-depth studies are required to fully identify the vesicular content and to understand how OMVs mediate antibiotic-resistance mechanisms. In fact, an investigation into the possible role played by OMVs in antibiotic-resistance mechanisms would open the door for an opportunity to develop new, innovative therapeutic strategies to fight antibiotic resistance in the “post-antibiotic era”.

## Figures and Tables

**Figure 1 antibiotics-12-00744-f001:**
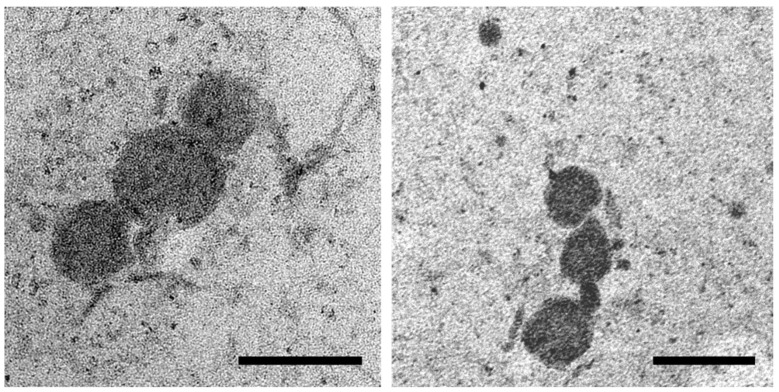
Representative transmission electron microscopy (TEM) images of OMVs isolated from *S.* Infantis culture. Scale bar 200 nm.

**Figure 2 antibiotics-12-00744-f002:**
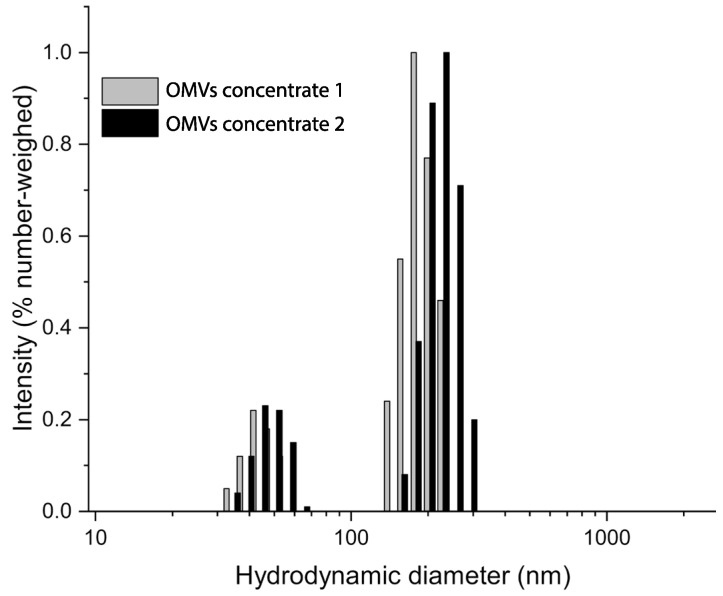
DLS analysis of OMVs concentrates 1 and 2. Samples were tested at 20 °C in a 0.22 µm filtered PBS medium. The hydrodynamic size distributions show two nearly overlapping populations at about 50 and 150 nm indicating size similarity of the OMV samples tested.

**Figure 3 antibiotics-12-00744-f003:**
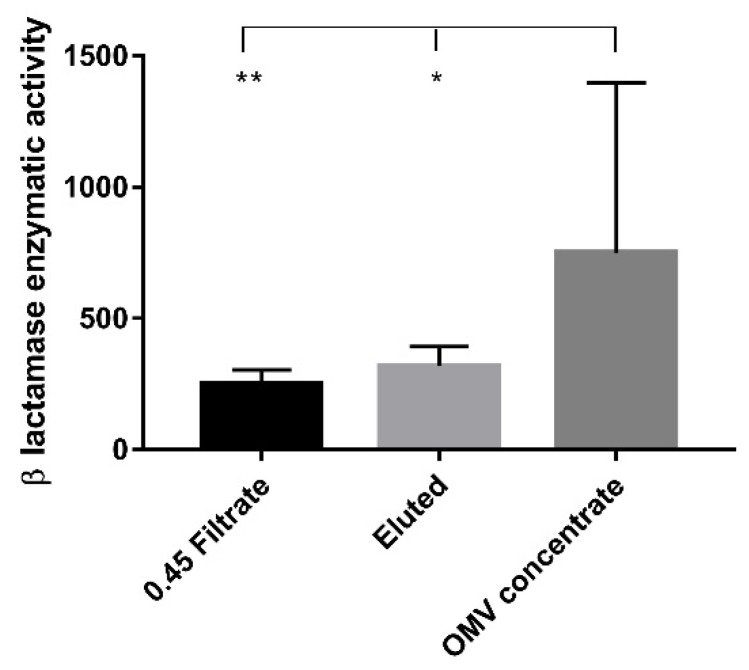
β-lactamase enzymatic activity. The bar graph shows the β-lactamase enzymatic activity expressed in milliunits/milligram of protein (mU/mg) measured in the three experimental conditions. Each value was normalized to the protein content and expressed as means ± standard deviation. Asterisks indicate significant differences in the β-lactamase activity. * *p* < 0.05, ** *p* < 0.01.

## Data Availability

Not applicable.

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
