# Peer review of "Evaluation of β-Lactamase Enzyme Activity in Outer Membrane Vesicles (OMVs) Isolated from Extended Spectrum β-Lactamase (ESBL) Salmonella Infantis Strains"

_antibiotics, 2023, doi:10.3390/antibiotics12040744_

Round 1
Reviewer 1 Report
The manuscript analyses the presence of Outer membrane vesicles in S. infants isolates by means of Electron Microscopy and other tests., focusing on the localization and the role of delivery of betalactamases enzymes. The research is very interesting and innovative.
Several comments could improve the quality of the manuscript.
line 20: Salmonella infantis ( salmonella in extensor).Please verify in guidelines if it is necessary for the first time to write Salmonella enterica subs. enterica
line 20: better to write resistant to beta-lactamases ( here and along the manuscript)
lines 54-68. please synthesize , the research is focused on AR
line 73-75: please re-write this sentence
line 100: please write ESBL producers , here and along the manuscript
line 103:with reduced resistance to.. this is not clear, please explain
line 108: S. infantis
line 115:do you mean cultures?
line 160: I'm not sure that intelligent is appropriate, please check
line 161: several instead of numerous
line 16 AR
Line 169-170 : rewrite ( no : It has also observed)
line 180: to instead of toward
line 180: how the isolates were found to be resistant to tetracycline etc( MIC, genotypic methods , etc)? please specify in MM.
line 184: delete 'for the first time'
line 225: it should be better to refer to WHO list of antimicrobials. ( critically important)
line 234: as Antibiotic resistance, not antimicrobial, is used in other parts, please put AR
line 241: isolates instead of strains
line 243: please provide more details about the typing
line 256, 262 : please add the company of the media
lines 303-304: the description of the test is too detailed, please synthesize
Conclusion:
please avoid "we" and use neutral form.
Reviewer 2 Report
The manuscript was well written.
My only concern was there must me mode detailed discussion.
Also, limitations of this study are missing.
Reviewer 3 Report
OMV’s can be used in biotechnological applications that require delivery of biomolecules such as vaccines. This manuscript by Toppi et al isolate the OMVs from S. Infantis β-lactam resistant strains and investigate their β-lactamase enzymes activity during their biogenesis. The findings are interesting, and manuscript is well presented. One of the major strengths of this findings, isolate OMV’s first time from MDR S. Infantis and measured the β-lactamase enzyme activity to understand whether β-lactamase enzymes are included in vesicles during their biogenesis. The authors might consider revising the manuscript, however, there are few comments given below.
How author measured the OMV production? If measured, give SDS-PAGE as supplementary file.
Can author clear/specify the envelope integrity?
Please antibiotic-resistance profile gives as supplementary file of S. Infantis.
Suggest providing GenBank access for sequence in the text.
All scientific names should be in Italic form in the text.
